# Using High-Frequency Information and RH to Estimate AQI Based on SVR

**DOI:** 10.3390/s21113630

**Published:** 2021-05-23

**Authors:** Jiun-Jian Liaw, Kuan-Yu Chen

**Affiliations:** Department of Information and Communication Engineering, Chaoyang University of Technology, 168, Jifeng E. Rd., Wufeng District, Taichung 413310, Taiwan; jjliaw@cyut.edu.tw

**Keywords:** AQI, visibility, digital image processing, SVR, high frequency information, RH

## Abstract

The Environmental Protection Administration of Taiwan’s Executive Yuan has set up many air quality monitoring stations to monitor air pollution in the environment. The current weather forecast also includes information used to predict air pollution. Since air quality indicators have a considerable impact on people, the development of a simple, fast, and low-cost method to measure the AQI value is a worthy topic of research. In this study, a method was proposed to estimate AQI. Visibility had a clear positive relationship with AQI. When images and AQI were compared, it was easy to see that visibility decreased with the AQI value increase. Distance is the main factor affecting visibility, so measuring visibility with images has also become a research topic. Images with high and low PM_2.5_ concentrations were used to obtain regions of interest (RoI). The pixels in the RoI were calculated to obtain high-frequency information. The high-frequency information of RoI, RH, and true AQI was used for training via SVR, which was used to generate the model for AQI estimation. One year of experimental samples was collected for the experiment. Two indices were used to evaluate the performance of the proposed method. The results showed that the proposed method could be used to estimate AQI with acceptable performance in a simple, fast, and low-cost way.

## 1. Introduction

In order to improve air quality, governments have established air quality standards. The Air Quality Index (AQI) in Taiwan is stipulated by the World Health Organization (WHO) [1]. It is divided into six categories (as shown in Table 1), which indicate the impact of air quality on health [2]. To measure AQI, the instrument first detects the concentration of pollutants, such as PM_2.5_, PM_10_, NO_2_, O_3_, SO_2_, and CO [3,4]. The impact of these pollutants on the human body is calculated as a sub-indicator. The maximum value of the sub-indicators is set as the current AQI. The measuring methods of the sub-indicators are different, as shown in Table 2 [2]. It is shown that the measurement procedures for the sub-indicators are different. Some methods are based on several hours of measurement results, while others are immediate measurements.

The government placed air pollution measuring instruments at air quality monitoring stations to monitor air quality in various regions. The values measured by the stations are accurate, but the instruments are expensive. Taiwan had established a total of 76 air quality monitoring stations before 2019 [5]. The cost of these monitoring stations is close to NT$10 million, and the annual maintenance and calibration costs are NT$700,000 to NT$800,000 [6]. In addition to the prohibitively high cost, there are also other shortcomings. For example, the range that can be measured is limited, the equipment needs to be calibrated regularly, and professional operators need to be trained. Therefore, developing a simple, fast, and low-cost method to measure the AQI value is an interesting topic of research.

PM_2.5_ and PM_10_ are prominent among the factors affecting air quality. Previous studies have pointed out that PM_2.5_ [7], PM_10_ [8], and relative humidity (RH) [9] have a serious impact on visibility [10,11]. It is not difficult to see that visibility has a positive relationship with AQI. Visibility is measured by trained professionals in the traditional way [12]. In recent years, various techniques for estimating visibility via images were developed [13,14]. The features, such as transmittance [15], entropy [16], contrast [17], and high-frequency information [18], are extracted from images and used to estimate visibility. The image characteristics of the scene, such as sky color, contrast, and spectral slope, are also used to estimate AQI [19,20]. In addition to image features, weather factors (such as RH, wind speed, or temperature) are used to adjust the parameters. In order to classify AQI into six categories, the multi-core learning method was applied. The results show that the average classification accuracy rate was 84%. 

In addition to the fixed scene, sky information of different scenes is also used to estimate AQI [21]. The sky area of the image can be used to classify AQI into six categories via the Gabor filter and K-NN classifier after calculating the transmittance [22]. In an experiment using 3422 images (divided into training and testing sets), an 82% classification accuracy was shown in result with cross-validation [23]. Under the premise that the main factor affecting visibility is distance [24], it is not easy to use sky information to establish an air pollution measurement method. This is because the distance from the sky cannot be measured. However, the sampled images for the related study were not sufficient. It is better to take sampling images up for one year to estimate the performance of the method.

Since distance is the main factor affecting visibility, the measurement of visibility with buildings at different distances has also become a research topic. Buildings with different distances or different features in the image can be divided into different regions. The region used to estimate the visibility or air pollution index are selected as regions of interest (RoI) [24,25]. The features (the distance, fog model, entropy, contrast, sky color, sky gradient, and sunlight incidence angle) in the RoI are compared with the air pollution index, and the coefficient of determination (R^2^) is obtained with acceptable results. Moreover, the pixel values are shown as having an exponential relationship with the different distance objects [26]. The RoI has achieved good results with respect to estimating air pollution with image processing technologies. However, too many features and manually selected RoI are disadvantages of the operation.

In order to solve the method of manually selecting RoI, the method of automatic selected RoI was proposed [27]. Two images of high and low PM_2.5_ concentrations were put into a group. The differences in the features of the group were calculated. After calculating serval groups of images, the area most affected by the changes in PM_2.5_ concentration is the RoI. The pixels in the RoI were used to extract high-frequency information such as the parameters for estimating the PM_2.5_ concentrations. The top three areas with the largest differences were compared, and the area with the largest differences had the best effect on PM_2.5_ concentration estimation. However, the experimental data excludes RH, and the method is not applied to estimate AQI.

In this study, a method of estimating AQI is proposed and is simple, fast, and affordable. Images with high and low PM_2.5_ concentrations are used to obtain the RoI. The pixels in the RoI are calculated to obtain high-frequency information. Then, SVR is applied to estimate AQI with high-frequency information and RH. The experiment included one-year data collection, and performance is indexed with R^2^ and RMSE.

## 2. Automatic Selection of RoI

If images are shot from the environment and compared with the AQI, it can be observed that when visibility decreased, AQI values increased. Figure 1 shows the images taken by a fixed camera. According to the observation of the fixed building (red frame) in the image, when the AQI changed from 25 to 170, the details of the building changed from clear to blurry. The manifestation of such visibility changes in the frequency domain is a reduction in high-frequency information. This is the main reason for using high-frequency variation changes to obtain the RoI.

A flow chart of the automatic selection of RoI is shown in Figure 2 [27]. To find the difference between high and low visibility images, two images with different PM_2.5_ concentrations (high and low) were treated as a group. Each image of a group had high-frequency information extracted individually. High-visibility images would have more high-frequency information than low-visibility images. The binarization method and morphology were applied to obtain pixels with large differences in high-frequency information within individual images. The XOR operation leaves pixels of two images with large high-frequency differences. The pixels with large differences were combined with neighbors into a region via labeling. The labeling process was conducted by an algorithm that gave connected pixels the same label. Pixels with the same label could be treated as the same region. In regions obtained by calculating every group, the region with the largest high-frequency information is the RoI.

## 3. To Estimate AQI with RH

Figure 3 is a flowchart of the proposed AQI estimation method. First, the RoI is selected after the images are inputted. Features are extracted from images in the RoI and calculate feature values, and then divide these values into training data and test data. The characteristic value, actual AQI value, and actual RH value in the test data are input to Support vector regression (SVR) [28] for training. The image of the test data is estimated through the trained SVR to estimate the AQI value. Afterward, the estimated AQI and the actual AQI will be evaluated through evaluation indicators. Finally, the estimated performance is obtained. In Section 3.1, the dataset used for the experiment is introduced. Since the RoI selected method was proposed by Liaw et al., the method of the RoI selection process will not be introduced here. The method used to calculate the feature values will be introduced in Section 3.2. Section 3.3 will introduce the training data and test data of the image. Section 3.4 will introduce the support vector regression. Section 3.5 will introduce the evaluation index for evaluating the performance of the proposed method.

### 3.1. Dataset

In this study, the images were taken from Taiwan’s air quality monitoring station as an image database. The Kaohsiung Renwu Monitoring Station was selected as it has relatively serious air quality issues compared to other areas of Taiwan. The images were taken for one year from August 2018 to July 2019, gathered from 7:00 a.m. to 17:00 p.m. One image was taken every 10 minutes, for a total of 21,720 images. The actual RH value and AQI value corresponding to these images were collected. Figure 4 shows the high AQI numerical image and the low AQI numerical image of the Renwu monitoring station in Kaohsiung, Taiwan. The images taken by the Kaohsiung Renwu Monitoring Station are images of urban buildings. The locations of the buildings range from far to near the shooting location. When the AQI value increases, the image pixels of distant buildings are more affected, and the contours and high-frequency features are gradually lost. Therefore, the image of the distant buildings was used. Figure 5 shows the distribution of AQI and RH values in the database. In the collected database, the distribution of AQI values was average. AQI values ranging from 15 to 160 had a considerable amount of image data. The RH value shows a normal distribution [29]. The RH value is very important in the experiment because the RH value was not included in the calculation of AQI, and RH will also affect visibility. Therefore, the RH value was added to the SVR to adjust the estimated AQI.

### 3.2. Value of Feature Calculation

When the AQI value increases, the outline of the distant target will gradually disappear, and the high-frequency information will gradually decrease. Therefore, the visibility of the scene was estimated by calculating the high-frequency information in the image. Here, the method of Sobel was used [30] to calculate the high-frequency information of the image. Sobel is a method of spatial domain filtering, which uses a high-pass filter. By performing horizontal and vertical convolution processing in the image [31] and adding the results of the horizontal and vertical convolution processing, the edge information of the image can be obtained. In the calculated Sobel images, higher pixels represent higher gradients. After the images are gray-scaled, two sets of 3 × 3 matrices are used to perform convolution processing on the gray-scaled image *A*. The convolution matrix for horizontal and vertical calculation is as follows, where *L_x_* represents the horizontal weight, and *L_y_* represents the vertical weight:(1)$$L_{x\;}=\left[\begin{array}{ccc} 1 & 0 & -1\\ 2 & 0 & -2\\ 1 & 0 & -1 \end{array}\right]*A,\;L_{y}=\left[\begin{array}{ccc} -1 & -2 & -1\\ 0 & \;0 & \;0\\ 1 & \;2 & \;1 \end{array}\right]*A$$

The horizontal and vertical gradients in the image pixels can be combined with the following formulas to calculate the magnitude of the gradient:(2)$$L_{x,y}=\left|L_{x}\right|+\left|L_{y}\right|$$

Figure 6 shows the horizontal and vertical gradient calculation of the image and the combination of the two images used to calculate the high-frequency information. *L_x_* in formula (7) calculates the horizontal weight, *L_y_* calculates the vertical weight, and then *L_x,y_* is calculated by formula (8) to calculate the high-frequency information.

After calculating the high-frequency information of the image, the next step was feature extraction. For RoI selection, the RoI selection method was proposed by Liaw et al. As shown in Figure 7, the feature can be obtained by calculating the pixel average of the high-frequency information in the RoI. The calculation formula is as follows:(3)$$Eigenvalues=\frac{1}{M\times N}\overset{M+p_{x}}{\underset{x=p_{x}}{\sum }}\overset{M+p_{y}}{\underset{y=p_{y}}{\sum }}G_{i}\left(x,y\right)$$
where *G_i_* is the pixel position (*x*,*y*) in the high-frequency information, the value of the feature is the average value of *G_i_* (*x*,*y*) in RoI, where the RoI is *M × N*. (*p_x_*, *p_y_*) is the coordinate position of the upper left corner of the RoI. Figure 8 shows the images presented at different AQI values, as well as the corresponding high-frequency images and feature values. When the AQI value increases, the image in the RoI gradually loses its outline and detailed texture. In the calculated high-frequency image, the high-frequency information of the building is also reduced. The calculated characteristic value also drops accordingly. Therefore, the value of AQI is negatively correlated with the calculated characteristic value of the high-frequency information.

### 3.3. Training Images and Testing Images

Before the experiment, the image data were divided into a training set and a test set. There was no duplicate data between the training set and test set. A cross-validation method was used to divide the data into the training set and test set. This method was used to randomly divide the data into a test set and training set at ratios of (1:9), (2:8), …, (8:2), (9:10). The collected database contained 21720 images. Table 3 shows the allocation table of the training and testing data. In the experimental results, the distribution ratios of these nine materials are shown.

### 3.4. Support Vector Regression

With the development of computer technology, there have been many technological breakthroughs in regression models [32]. Regression models are also used in the field of machine learning. The SVR application was used in this study. SVR is an application method used by the Support Vector Machine (SVM) [33] to solve regression estimation problems. The concept of SVM is shown in Figure 9. On two-dimensional data, the red and blue data are needed to divide, so they can be divided with a one-dimensional line. Obviously, Figure 9c has the best segmentation result because the segmentation line has the farthest distance from the data point. However, usually, the data is not only two-dimensional, so the data is needed to be mapped to a higher dimension to divide the data. Two-dimensional data can be divided by one-dimensional lines, and after the data is mapped to a three-dimensional space, the data can be divided by a two-dimensional plane. Therefore, when the data is mapped to an n-dimensional hyperplane, the data can be divided by an (n-1)-dimensional hyperplane.

The principle of SVR is the same as that of SVM, but the concept is different. Figure 10 shows the conceptual difference between SVM and SVR. SVM is used to find a hyperplane that separates the data, while SVR is used to find a hyperplane that can make all data points close to this hyperplane. When these data points are closer to the regression hyperplane, the estimated data will be more accurate.

Assuming that the system is linear, there are several planes in the feature space that can accurately express the relationship between the two types of data. There will be a classification boundary which can cause a small margin between the classifications. The best hyperplane should express different data on one hyperplane as much as possible. Taking a two-dimensional coordinate system as an example, the data set in space is composed of **x**_i_ and *y_i_*, and it can be written as:(4)$$\begin{array}{c} \left\{x_{i},y_{i}\right\},\;i=1,\ldots ,n\\ y_{i}\in \left\{-1,1\right\} \end{array}$$
where **x**_i_ is the data point, and *y*_i_ is the label of the classification. The hyperplane equation can be set as:(5)$$f\left(x\right)=w^{T}x_{i}+b=0$$
where *w* is the normal vector, 
and *b* is the offset. The SVR is used to find the solution of the 
following [34]:

Minimize $\frac{1}{2}∥w^{2}∥+C\sum_{i}\left(\xi_{i}+\xi_{i}^{*}\right)$
(6)$$\mathrm{Subject}\;\mathrm{to}\;\{\begin{array}{c} w\cdot x_{i}+b-y_{i}\leq \epsilon +\xi_{i}\\ y_{i}-w\cdot x_{i}-b\leq \epsilon +\xi_{i}^{*}\\ \xi_{i}\;\mathrm{and}\;\xi_{i}^{*}\geq 0 \end{array}$$
where ||.|| is the Euclidean norm, ε represents the allowable error, C is the tradeoff between the flatness and the allowable error, and $\xi_{i}\;\mathrm{and}\;\xi_{i}^{*}$ are the *i*th slack variable. Lagrange Multiplier is used to solve the above optimization problem [35].

### 3.5. Evaluation Index

The correlation coefficient is used to measure the degree of correlation between two pieces of data. R^2^ [36] and root-mean-square error (RMSE) are used as the evaluation standard for estimating AQI, and the calculation formula of R^2^ is as follows:(7)$$R^{2}=1-\frac{\sum_{i=1}^{n}{\left(a_{i}-b_{i}\;\right)}^{2}}{\sum_{i=1}^{n}{\left(a_{i}-{\bar{b}}_{i}\right)}^{2}}$$

Among them, $a_{i}$ is the *i*th estimated value, $b_{i}$ is the *i*th actual value, ${\bar{b}}_{i}$ is the average value of ${\;b}_{i}$, and n is the number of data. R^2^ is between 0 and 1. A value closer to 1 represents the estimated data, which is more linear. That is to say, when R^2^ is 1, this means the estimated AQI value is exactly the same as the actual AQI value. Generally speaking, when R^2^ is greater than 0.5, it means that the two pieces of data have a greater correlation. 

The calculation formula of RMSE is as follows:(8)$$RMSE=\sqrt{\frac{1}{n}\sum_{i=1}^{n}{\left(a_{i}-b_{i}\right)}^{2}}$$

RMSE represents the root mean square error between the estimated value and the actual value. If the difference between the estimated AQI value and the actual AQI value is smaller, the RMSE will be smaller. Therefore, when the RMSE is closer to 0, it means that there is no error in the actual value.

## 4. Experiment

In this section, the results of the entire experiment are introduced. There were some factors that could affect the results of the experiment, so we needed to exclude invalid data. Details will be introduced in Section 4.1. Section 4.2 shows all the experimental results, showing the AQI estimation performance under different training and test allocation ratios. Section 4.3 discusses the experimental results and how future experiments can improve these problems.

### 4.1. Exclude Invalid Data

In the collected image database, images were taken every 10 min, but the AQI value was one image per hour. This resulted in a situation where there is only one AQI value but six images in an hour. This is because when the air quality monitoring station measures the AQI value, it takes about 1 hour of sampling time. However, in actual situations, the feature values calculated from the six images within this hour are likely to show large changes. This is because air pollutants are usually not uniform in the air, and visibility levels within an hour may vary greatly due to wind speeds, weather conditions, and relative humidity of the day. Therefore, when the variation of the characteristic value within one hour is too large, the estimated AQI will also have big differences. However, the AQI corresponding to these six sheets has only one value. The performance evaluated at this time would be affected. As shown in Figure 11, the AQI in this hour had only one value, but the characteristic value of the six images changed from 17.5 to 40.7. At this time, the estimated AQI value would also show a very large change. Therefore, we excluded data whose characteristic values changed too much.

The data with characteristic value changes excessively within one hour were also excluded. First, the feature values of the images in the training set were calculated. The characteristic value Standard Deviation(SD) [37] was calculated from the six characteristic values per hour and the mean SD from the standard deviation of each hour’s characteristic value in the database. In the experiment, a threshold value should be set to exclude data with characteristic value SDs that are too high. As shown in Figure 12, data were treated with hourly feature value standard deviations greater than the mean SD as invalid data and excluded. The calculated mean SD was 3.66. When excluding data larger than the mean SD, 35% of the data of the appointment was excluded. It is shown that this 35% will affect the experimental estimates. The dataset after excluding invalid data includes 13,945 images. The performance before and after excluding invalid data is compared in Section 4.2.

### 4.2. Experimental Result

The high-frequency feature value, RH, and actual AQI value of the image were added to the SVR, and the AQI values of the images in the test set were estimated. R^2^ and RMSE were used to evaluate the performance before and after excluding invalid data. Table 4 shows the AQI estimation results of the training to test ratio from [1:9] to [9:1], and Figure 13 shows a comparison chart of the AQI estimation results.

Each evaluation value in Table 4 is the average result of three times experiments. The average experimental results before excluding invalid data obtained R^2^ of 0.646 and RMSE of 23.5. After excluding invalid data, the R^2^ was 0.678 and the RMSE was 22.3. We used experiments to prove that the exclusion of invalid data indeed affected the estimation results, and these data cannot accurately estimate the AQI value.

After excluding invalid data, the training to test ratio has the best estimation performance at [7:3]; R^2^ was 0.694 and RMSE was 21.8. The estimated performance when the ratio of training to test was (1:9) was the worst; the R^2^ was 0.655 and RMSE was 23.1. This can be seen in Figure 14. The ratio of training and testing starts from (1:9) to (7:3) to reach the peak of estimated performance, and when it reaches (8:2) and (9:1), the estimated performance begins to decline. In particular, the worst and best-estimated performance difference R^2^ was only about 0.04, and for RMSE is was only about 1.3. Figure 15 shows the relationship between the best performance estimate and the actual AQI. It was found that the data was quite concentrated. There were better estimation results when the AQI was lower and higher. When the AQI value was medium, the estimated performance was worse.

### 4.3. Discussion of Experimental Results

In previous studies, linear or non-linear regression models were mostly used, so the RH weather characteristics could not be used to adjust the estimation results. Data with RH that are too high would be excluded at this time. In the experiment, SVR was used to estimate AQI, and RH weather characteristics were added to adjust the estimated AQI value. This can solve the impact on visibility when the RH was too high. The method of automatically calculating RoI was also used. This method can better find the area suitable for estimating AQI. Therefore, the proposed method has better performance to estimate the AQI value.

In the experimental results, the estimated AQI performance R^2^ reached a correlation coefficient of 0.694. The worst R^2^ was 0.655. In fact, the estimated performance was not particularly good. There are several factors that could cause estimated performance to decline. The first indicators that have a greater impact on visibility are PM_2.5_, PM_10_, and RH. However, in addition to these three indicators, there will be other factors that affect visibility. At the same time, light and weather could also affect visibility, so it was estimated that performance will decrease.

Although PM_2.5_ and PM_10_ affect the AQI value to a large extent, other secondary indicators also affect the AQI value to a lesser extent. In other words, it was difficult to estimate 100% correct AQI values.

In this experiment, only the high-frequency information of the image was used. However, the method of estimating visibility through images was not limited to high-frequency image information. Other experiments often use methods such as image transmittance, entropy, and contrast to estimate visibility. Moreover, SVR is a very powerful regression model, which can input quite a lot of features to perform regression operations. Therefore, estimating AQI through more features can improve estimation performance and should be the focus of future works.

Although the proposed method could estimate the effective AQI value, this method still has some limitations. First, it requires a lot of data for training. In the experiment, the highest estimation performance was achieved when the ratio of the training data to test data was 7:3. Currently, about 9800 images were used for training. Training with these images requires actual AQI values and actual RH values. In addition, the scene and angle of the photos cannot be changed. Therefore, when the scene target in the RoI changes, the RoI is needed to recalculate. Finally, the proposed method is only suitable for daytime images. The AQI value for the images cannot be estimated at night. In future experiments, in addition to improving the performance of estimating AQI, these limitations with respect to estimating AQI will be resolved.

## 5. Conclusions

In this study, a method was proposed to estimate AQI. The proposed method included applying an automatically selected RoI obtained from the difference between images with high and low visibility. The high-frequency information of RoI, RH, and true AQI was used to train via SVR, which was used to generate the model of estimation. One year of experimental samples, which included 21,720 images with corresponding RH and AQI, were collected from a fixed monitoring station. According to the results, the performance of AQI estimation could reach R^2^ = 0.694 and RMSE = 21.8 when the training-to-test ratio was 7:3. The experiments showed that including RH did not reduce the performance in a whole year of sampled data. The proposed method could be used to estimate AQI with acceptable performance in a simple, fast, and low-cost way. Since the image features (rather than chemical composition) were used to estimate AQI, the correlation between features and AQI is worth exploring in the future. In addition, SVR is a kind of machine learning method, so the question of how many input effective features can be used to improve the performance should also be a focus of future efforts.

## Figures and Tables

**Figure 1 sensors-21-03630-f001:**
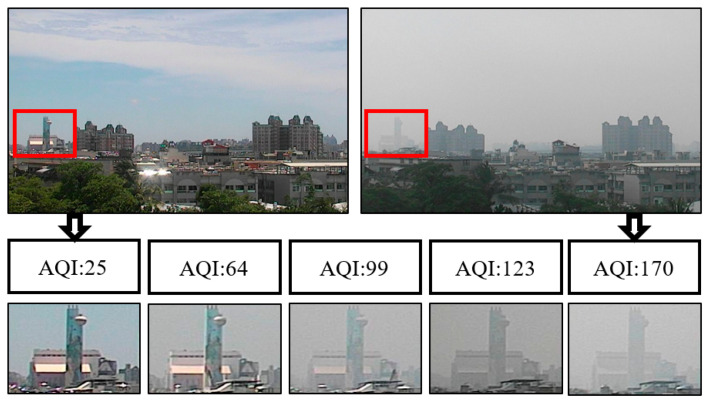
A schematic diagram of AQI and visibility changes.

**Figure 2 sensors-21-03630-f002:**
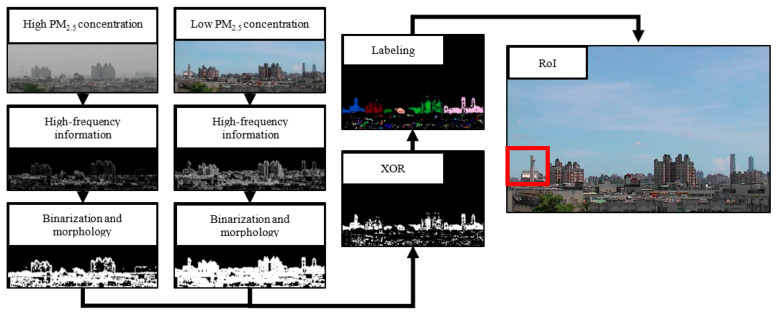
Automatic selection of the RoI flowchart.

**Figure 3 sensors-21-03630-f003:**
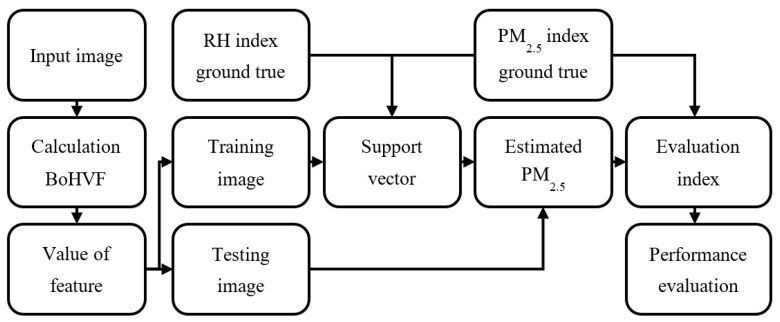
A flow chart of the method of estimating AQI.

**Figure 4 sensors-21-03630-f004:**
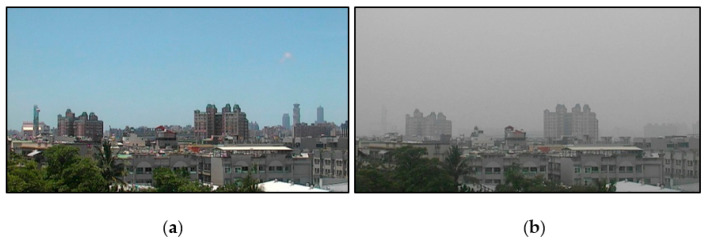
Images from the Renwu monitoring station in Kaohsiung, Taiwan; (**a**) low-AQI numerical image and (**b**) high-AQI numerical image.

**Figure 5 sensors-21-03630-f005:**
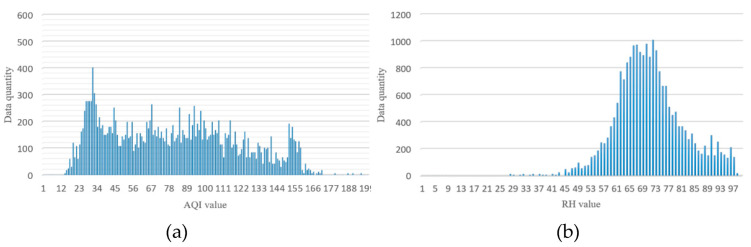
The distribution of AQI and RH values at Renwu Monitoring Station in Kaohsiung, Taiwan; (**a**) AQI value distribution and (**b**) RH value distribution.

**Figure 6 sensors-21-03630-f006:**
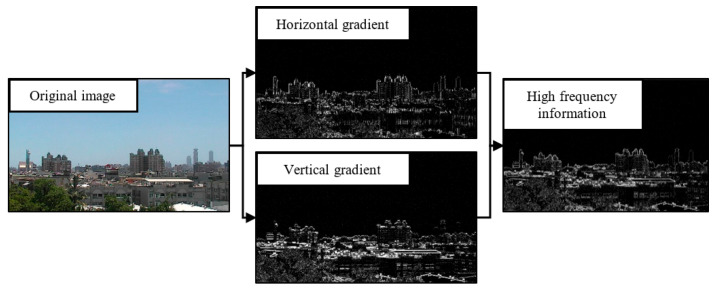
The image performs horizontal and vertical gradient calculations and calculates the high-frequency information.

**Figure 7 sensors-21-03630-f007:**
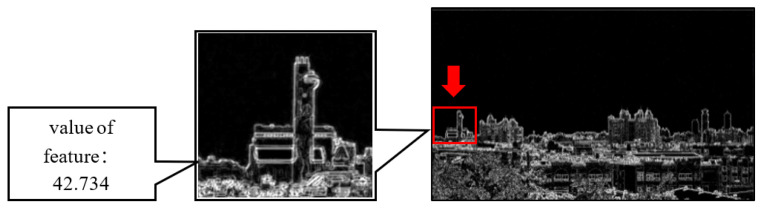
The method for calculating the feature value in the RoI.

**Figure 8 sensors-21-03630-f008:**
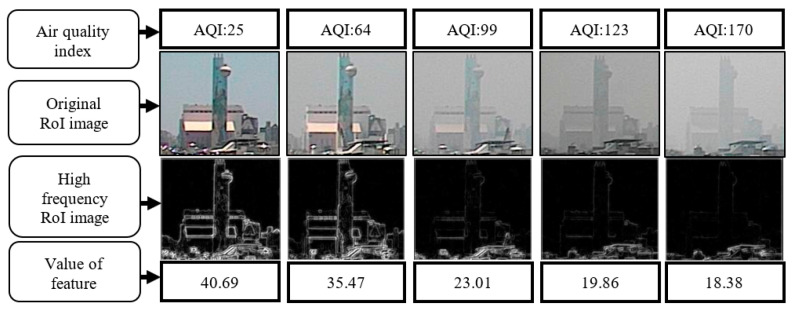
High-frequency images and feature values corresponding to different AQI values.

**Figure 9 sensors-21-03630-f009:**
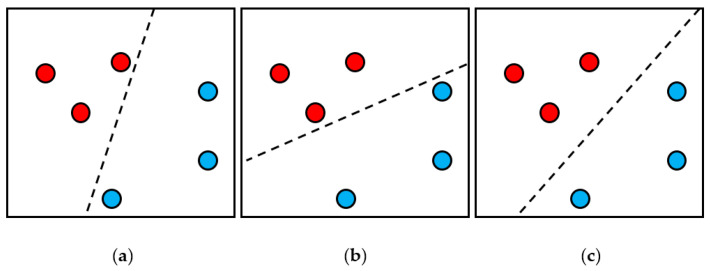
A conceptual diagram of the SVM segmentation data; (**a**,**b**) show the poor segmentation results, (**c**) shows the best segmentation results.

**Figure 10 sensors-21-03630-f010:**
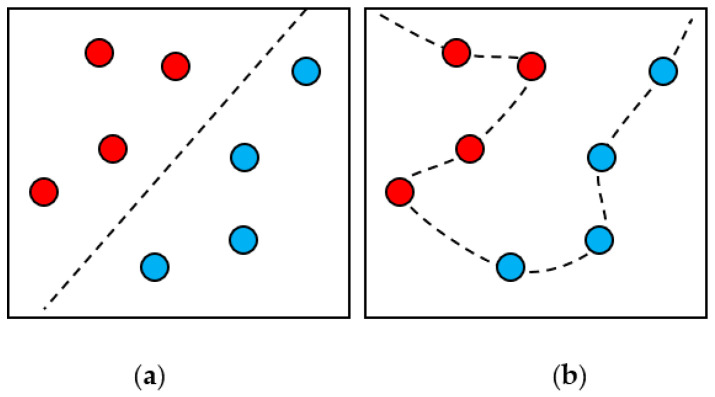
The conceptual difference between SVM and SVR; (**a**) the dividing line is far away from the data point; (**b**) the dividing line is close to the data point.

**Figure 11 sensors-21-03630-f011:**
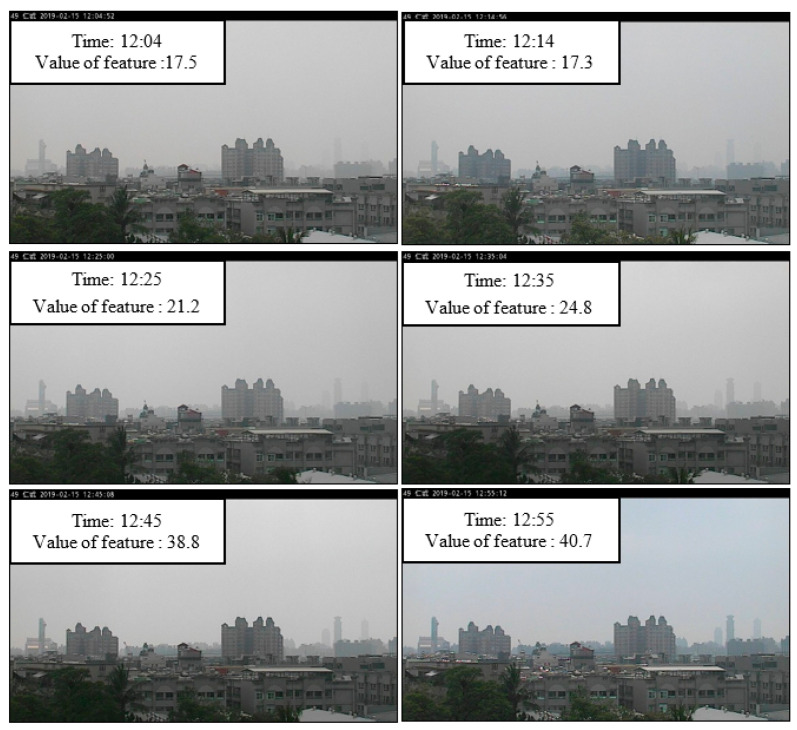
Value of feature change within one hour.

**Figure 12 sensors-21-03630-f012:**
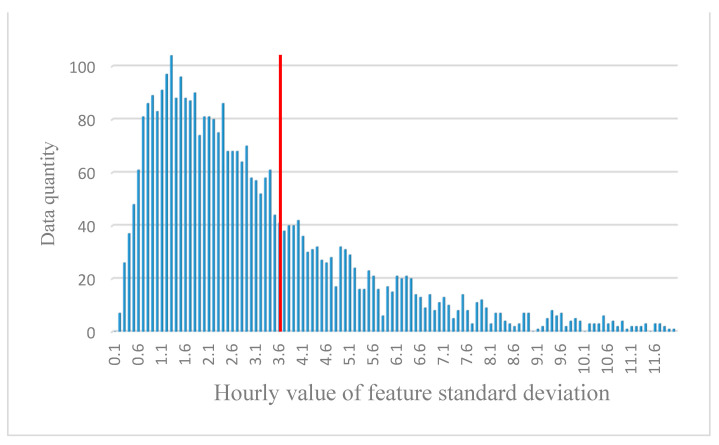
The standard deviation distribution of hourly feature values.

**Figure 13 sensors-21-03630-f013:**
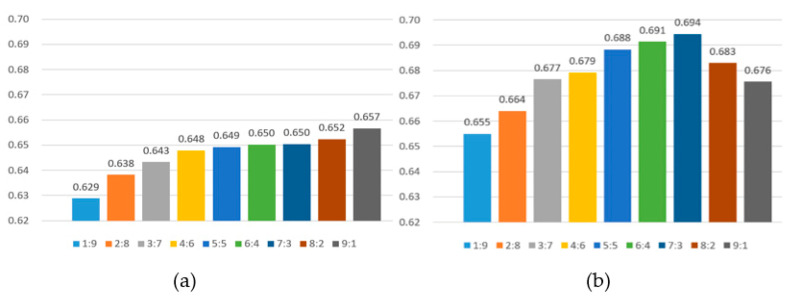
The R^2^ of the estimated AQI and the actual AQI values; (**a**) invalid data is not excluded; (**b**) invalid data is excluded.

**Figure 14 sensors-21-03630-f014:**
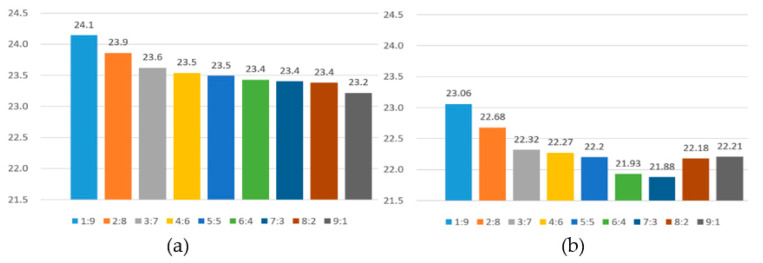
The RMSE of the estimated AQI and the actual AQI value; (**a**) invalid data is not excluded, (**b**) invalid data is excluded.

**Figure 15 sensors-21-03630-f015:**
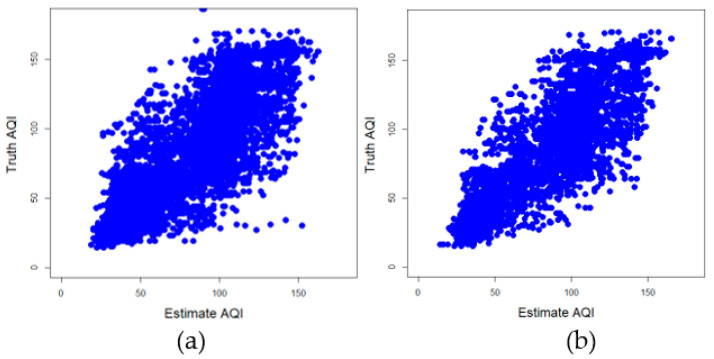
The relationship between the estimated and actual AQI when the ratio of training-to-test is 7:3; (**a**) invalid data is not excluded, (**b**) invalid data is excluded.

**Table 1 sensors-21-03630-t001:** AQI categories and the corresponding human health effects [2].

AQI	Concern Levels	Color	Description
0–50	Good	Green	Air quality is good; little air pollution or no risk.
51–100	Moderate	Yellow	Air quality is acceptable. There may be a risk for some people who are sensitive to air pollution.
101–150	Unhealthy for sensitive groups	Orange	Sensitive people may experience health effects. The general public is less likely to be affected.
151–200	Unhealthy	Red	The general public may experience health effects; sensitive people may experience more serious health effects.
201–300	Very unhealthy	Purple	The risk of health effects is increased for everyone.
301–500	Hazardous	Dark red	Everyone is more likely to be affected.

**Table 2 sensors-21-03630-t002:** AQI’s sub-indicator calculation method [2].

Sub-Indicator	Calculation Method
O_3, 8h_	moving average value of the last 8 h
O_3_	real-time value
PM_2.5_	0.5 × average of the first 12 h + 0.5 × average of the first 4 h
PM_10_	0.5 × average of the first 12 h + 0.5 × average of the first 4 h
CO	moving average value of the last 8 h
SO_2_	real-time value
SO_2, 24h_	average in the last 24 h
NO_2_	real-time value

**Table 3 sensors-21-03630-t003:** Distribution of the training data and test data.

Training:Testing	Training Data	Test Data
1:9	2172	19,548
2:8	4344	17,376
3:7	6516	15,204
4:6	8688	13,032
5:5	10,860	10,860
6:4	13,032	8688
7:3	15,204	6516
8:2	17,376	4344
9:1	19,548	2172

**Table 4 sensors-21-03630-t004:** AQI estimated performance table.

	Invalid Data not Excluded		Exclude Invalid Data	
Training: Testing	R^2^	RMSE	R^2^	RMSE
1:9	0.629	24.1	0.655	23.1
2:8	0.638	23.9	0.664	22.6
3:7	0.643	23.6	0.676	22.3
4:6	0.648	23.5	0.697	22.2
5:5	0.649	23.5	0.688	22.2
6:4	0.650	23.4	0.691	21.9
7:3	0.650	23.4	0.694	21.8
8:2	0.652	23.4	0.683	22.1
9:1	0.657	23.2	0.676	22.2
Average value	0.646	23.5	0.678	22.3

## Data Availability

Not applicable.

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
