# Peer review of "Using High-Frequency Information and RH to Estimate AQI Based on SVR"

_sensors, 2021, doi:10.3390/s21113630_

Round 1
Reviewer 1 Report
The study is not clear enough in the description of motivation, the exclusion invalid data are not detailed, and the machine learning method is not described in detail.
Author Response
The reply content is as document "Response for Comments_Reviewer_1"

Reviewer 2 Report
The manuscript proposes and assesses a method to estimate air quality indicators through images and imaging processing methods.
The experimental results show that the proposed method can effectively estimate the AQI value.
The manuscript is well structured, balanced, but in terms of writing it should be improved, perhaps a review by a native speaker would be important.
The methodology is clear but since they are using SVM, the configuration and kernel type is not disclosed and the results presented are insuficient.
Authors should refrain from using personal pronouns such as "we" and "our" throughout the text and I encourage them to write it in an impersonal form of writing.
Line 79: "We think that...", this expression is vague and should be avoided, science is not based in conceptual thoughts.
At the results, since classification is involved, not confusing matrix, accuracy, recall or F-Score are provided. Relying only in R2 is not enough.
Discussion section must also be extended and related with previous research and the study limitations should be described.
Only 6 out of 30 references are of the last 5 years, some updated references are required.
Author Response
The reply content is as document "Response for Comments_Reviewer_2"

Reviewer 3 Report
Dear Editor and Authors,
The MS entitled: Using high-frequency information and RH to estimate AQI based on SVR submitted to the MDPI journal “Sensors”, presented a method to estimate air quality indicators through image processing using the images from the air quality monitoring stations. The authors presented the algorithms of image transformations and estimation procedure. Although, the paper contains the interesting concept of air pollution monitoring, the way how it is written doesn’t allow clear understanding. The authors should first restructure the paper to follow the proper content of each paragraph. The introduction is completely incorrect, it already contains part of the results and discussion! In the paper there is a lot of repetition of the same sentences written in different ways, what makes the outcome very confusing. It is very difficult to follow the logic of the procedure as it is not ordered in a proper sequence. It gives an impression of chaos. I believe this paper can be an interesting submission but it requires a careful restructuring, and totally rewritten by focusing on the clear ordered steps and conclusions from them, prior to any serious review. Paper needs also a substantial English language correction.
Author Response
The reply content is as document "Response for Comments_Reviewer_3"

Round 2
Reviewer 1 Report
I recommend to accept the paper
Author Response
Thank you very much for the reviewer's suggestions on our research.
These suggestions are very helpful to us.
Reviewer 2 Report
The authors have significantly improved the manuscript addressing the reviewers comments and suggestions.
However, I am still not happy in authors using personal pronouns at the text such as "We" and "our", you are the authors of the work, it is inelegant to use this scientific writing. I advise you to use impersonal writing instead.
Author Response
Thank you very much for the reviewer's suggestions on our research.
These suggestions are very helpful to us.
The reply content is shown in the attachment.

Reviewer 3 Report
Dear Authors,
Thanks to the authors for following my comments. The article has been ordered and enriched with missing content. The level of the MS has been significantly improved. The new structure of the article allows understanding.
Author Response

(The authors gave the same response as above.)
